# Behaviour Change Techniques Used in Mediterranean Diet Interventions for Older Adults: A Systematic Scoping Review

**DOI:** 10.3390/nu15051189

**Published:** 2023-02-27

**Authors:** Ashlee Turner, Haley M. LaMonica, Victoria M. Flood

**Affiliations:** 1Sydney School of Health Sciences, Faculty of Medicine and Health, The University of Sydney, Sydney, NSW 2006, Australia; 2Translational Research Collective, Faculty of Medicine and Health, The University of Sydney, Sydney, NSW 2006, Australia; 3University Centre for Rural Health, Faculty of Medicine and Health, The University of Sydney, Northern Rivers, Lismore, NSW 2480, Australia

**Keywords:** Mediterranean diet, interventions, behaviour change techniques, older adults, scoping review

## Abstract

Mediterranean diet interventions have demonstrated positive effects in the prevention and management of several chronic conditions in older adults. Understanding the effective components of behavioural interventions is essential for long-term health behaviour change and translating evidence-based interventions into practice. The aim of this scoping review is to provide an overview of the current Mediterranean diet interventions for older adults (≥55 years) and describe the behaviour change techniques used as part of the interventions. A scoping review systematically searched Medline, Embase, CINAHL, Web of Science, Scopus, and PsycINFO from inception until August 2022. Eligible studies were randomized and non-randomized experimental studies involving a Mediterranean or anti-inflammatory diet intervention in older adults (average age > 55 years). Screening was conducted independently by two authors, with discrepancies being resolved by the senior author. Behaviour change techniques were assessed using the Behaviour Change Technique Taxonomy (version 1), which details 93 hierarchical techniques grouped into 16 categories. From 2385 articles, 31 studies were included in the final synthesis. Ten behaviour change taxonomy groupings and 19 techniques were reported across the 31 interventions. The mean number of techniques used was 5, with a range from 2 to 9. Common techniques included instruction on how to perform the behaviour (*n* = 31), social support (*n* = 24), providing information from a credible source (*n* = 16), information about health consequences (*n* = 15), and adding objects to the environment (*n* = 12). Although behaviour change techniques are commonly reported across interventions, the use of the Behaviour Change Technique Taxonomy for intervention development is rare, and more than 80% of the available techniques are not being utilised. Integrating behaviour change techniques in the development and reporting of nutrition interventions for older adults is essential for effectively targeting behaviours in both research and practice.

## 1. Introduction

Older people make up a significant portion of Australia’s population, with one in six people being 65 years or older [1]. The number and proportion of older adults in the population is projected to continue to rise over the next 40 years, due to increasing life expectancy and declining fertility rates [1,2]. While a portion of this population will be in good health, completely independent, and active in their communities, many may experience more marked effects associated with ageing, including decreasing quality of life, illness, and disease [3]. However, evidence suggests poor health in later life is not inevitable, with recent research showing that many ill-health characteristics associated with ageing can be attributed to modifiable lifestyle factors, such as hypertension, hypercholesterolemia, diabetes, obesity, and low levels of physical exercise [4].

Diet is a key lifestyle behaviour that is fundamental to overall health and quality of life. Indeed, adequate nutrition plays a role in the prevention and management of many chronic illnesses that affect older adults [5]. Despite the growing evidence demonstrating the role of diet in chronic disease management and healthy ageing, epidemiological and population evidence suggests that dietary quality tends to be poor among the older population, characterized by low intakes of fruits, vegetables, and wholegrains, alongside increased consumption of refined grains, sugars, and red and processed meats [6,7,8,9]. Dietary patterns such as the Mediterranean diet protect against age-related chronic conditions, including cardiovascular disease (CVD), type 2 diabetes, metabolic syndrome, risk of cancer, and reduced cognitive functioning [10]. While a Mediterranean-style diet is often endorsed in practice, adherence is generally low to moderate at best among older adults [11], which is unsurprising given the reports regarding diet quality in this group and the complexities associated with changing health behaviours [12]. While there is evidence from intervention programs that older adults can adhere to a Mediterranean diet over a short period [13,14], the long-term effectiveness, and therefore their capacity to change behaviour, of such programs is not well established. Thus, there is a need to identify and understand the effective components of behavioural interventions that contribute to increased dietary adherence and promote sustained health behaviour change in the long term.

Behaviour change theories provide a foundation to explain why people adopt a behaviour. Interventions that are theory-informed, particularly those that are underpinned by theories such as the social cognitive theory, theory of planned behaviour, and the transtheoretical model, have the potential to be more effective at promoting behaviour change and improving health outcomes than those developed in the absence of a guiding theory [15]. However, theory alone offers little explanation on the specific elements that contribute to the initiation and maintenance of behaviours [16]. Indeed, behavioural interventions are often complex and contain many interrelated components, which makes them difficult to replicate in research, adopt in real-world settings, and synthesize in systematic literature reviews [17,18]. Recent advancements in behaviour change science have led to the development of the Behaviour Change Technique Taxonomy, which provides a standardized classification system for behaviour change techniques to help develop new interventions, as well as to facilitate the replication of existing interventions [17]. Understanding the “active” ingredients of interventions is a crucial step in translating evidence-based interventions into practice. However, the ability to do this is contingent on the body of available evidence [17].

This review identifies and summarizes Mediterranean diet intervention studies in older adults. Specifically, we summarize the outcome measures related to diet components used in these studies (e.g., diet adherence, food and nutrient intake, nutrient biomarkers) and describe the delivery format and behaviour change strategies used in the intervention. Describing the behaviour change strategies used in diet interventions is an essential first step for identifying and understanding the “active” ingredients that produce the desired behavioural changes. The purpose of this review is to identify and describe behaviour change techniques used in Mediterranean diet interventions for older adults. To our knowledge, no scoping reviews have been conducted on Mediterranean diet interventions and behaviour change in older adults.

## 2. Methods

This review followed a systematic approach, with a protocol prospectively registered at Open Science Framework Registries (https://osf.io/28evm/, accessed on 18 February 2022). Search strategies, study selection, and result reporting were described according to the Preferred Reporting Items for Systematic Reviews and Meta-Analyses extension for Scoping Reviews (PRISMA-ScR) checklist (see Appendix A for PRISMA-ScR checklist) [19].

### 2.1. Search Strategy

The search strategy aimed to locate published studies. The databases searched included Medline, Embase, CINAHL Web of Science, Scopus, and PsycINFO. The search was not limited by date, beginning from inception and extending to August 2022. The search strategy was adapted for each database (see Table 1 for the search strategy for the Ovid MEDLINE database). The MeSH Keywords used across searches were ‘Diet, Mediterranean’, ‘Aged’, ‘Clinical Trials as Topic’, and ‘Randomized Controlled Trials as Topic’. Linked papers to the included studies (i.e., protocol papers and those presenting the results of other outcomes) were also collected. Reference lists of included studies were examined to identify additional eligible records.

### 2.2. Eligibility Criteria

Table 2 includes the detailed inclusion and exclusion criteria for selected articles. Briefly, articles were included if they reported on active Mediterranean or anti-inflammatory diet interventions in adults with an average age of 55 years and older. This age range was chosen due to the increasing prevalence and, in turn, burden of diet-related chronic diseases beginning from the age of 55 years. In cases with multiple publications arising from the same study, the paper with the most comprehensive description of the methodology, intervention, and results on dietary outcomes was used.

### 2.3. Study Selection and Data Extraction

Following the search, all identified articles were collated into EndNote X9 and imported into the Covidence systematic review software [20] for duplicate removal and review. The review process consisted of two phases of screening: (1) title and abstract review and (2) full-text review. In the first phase of screening, title, and abstracts for retrieved articles were assessed against inclusion criteria by two independent reviewers (AT and HL). In the second phase, full-text articles were scanned in detail again by two independent reviewers (AT and HL), and studies that did not meet inclusion criteria were excluded. Any disagreements that arose between the reviewers during either phase of the review process were resolved through discussion with a senior reviewer (VF), who approved the final list of included studies. The data extraction was conducted by one reviewer (AT) and then discussed among the research team to reach a consensus. This was completed using a data extraction tool developed in Microsoft Excel and included (1) study design, (2) sample characteristics and setting, (3) study objective, (4) description of the dietary intervention, (5) delivery methods and format of the intervention, (6) dietary outcome measures, and (7) study results.

### 2.4. Coding of Behaviour Change Techniques

The Behaviour Change Technique Taxonomy (version 1; BCTTv1) was used to assess the inclusion of behaviour change techniques in intervention content. The BCTTv1 describes 93 behaviour change techniques organized into 16 hierarchies and provides a standardized method for reporting techniques used in behaviour change interventions. Behaviour change techniques were coded by one reviewer (AT) after completing an online training course (http://www.bct-taxonomy.com/, accessed on 11 April 2022). The full text of included articles, additional secondary publications of the same intervention, and Appendix A or protocols were analysed and coded. In studies where multiple behaviours were targeted (e.g., diet and physical activity), only the behaviour change techniques specific to the diet component were coded.

Two assumptions were made when coding. Where interventions mentioned “education”, we coded 4.1 instruction on performing the behaviour and 5.1 information on health consequences. When “training” was mentioned, it was coded as 4.1. This approach was used to acknowledge a minimum of the educational strategies used in the interventions [21].

## 3. Results

### 3.1. Search Strategy Results

The flow of study identification and selection is outlined in Figure 1. The search strategy identified 2384 articles for screening, and 1 additional article was identified through a manual search/searching the reference lists of included articles. Following title and abstract screening, 256 papers were identified for full-text review, from which 225 did not meet inclusion/exclusion criteria and were excluded. A total of 31 articles were included in the current review.

### 3.2. Study Characteristics

The characteristics of the included studies are presented in Appendix A. The papers included in the review were published between 2005 and 2022. Studies and the associated levels of evidence [22] included randomized controlled trials (Level I) (*n* = 25) and non-randomized (Level II) (*n* = 6) study designs. Sample sizes ranged from *n* = 15 to *n* = 7447. Five studies included females only, and one study included males only.

The 31 studies were conducted in the USA (*n* = 8), Spain (*n* = 7), Australia (*n* = 6), United Kingdom (*n* = 1), Ireland (*n* = 1), Greece (*n* = 1), Italy (*n* = 1), Sweden (*n* = 1), Luxembourg (*n* = 1), Germany (*n* = 1), Taiwan (*n* = 1), and Iran (*n* = 1). There was one multi-country study conducted in Europe, which spanned across Italy, Poland, France, the Netherlands, and the United Kingdom.

Intervention length ranged from 4 weeks to 2 years. Only six studies included a longitudinal follow-up component after the initial intervention (19.4%), ranging from 6 weeks to 12 months post-intervention.

Nineteen interventions were single component, targeting diet only (61.3%). The remaining 12 were combined lifestyle interventions, targeting physical activity and diet (38.7%).

Most studies included participants with comorbidities common among older adults, including diagnosed CVD and high cardiovascular risk (*n* = 9), metabolic syndrome (*n* = 2), type 2 diabetes (*n* = 2), cognitive impairment (*n* = 2), hypertension (*n* = 1), rheumatoid arthritis (*n* = 1), knee osteoarthritis (*n* = 1), breast cancer (*n* = 1), prostate cancer (*n* = 1), Parkinson’s disease (*n* = 1), and heart or lung transplant patients (*n* = 1). Nine studies were conducted in reportedly healthy participants.

Comparison groups included usual or standard care (*n* = 12) that included the provision of typical information about dietary guidelines and general healthy eating (*n* = 6) or reducing dietary fat (*n* = 1). Two studies provided control participants with basic written instructions about the Mediterranean diet. Three interventions utilized the same structure for the control group as those in the treatment group but applied low-fat dietary principles. In six studies, participants were instructed to maintain their habitual diet (i.e., no dietary changes implemented). Three studies were combined lifestyle interventions, where the control participants engaged in the same Mediterranean diet intervention as the treatment participants (intervention differed in the provision of physical activity). Five studies were pre/post and did not include a control group.

### 3.3. Intervention Delivery Methods

Interventions were predominately delivered individually (*n* = 16, 51.6%), with four delivered in a group format (12.9%) and 10 delivered in a mixed individual and group format (32.3%). One study gave participants the option of completing the intervention in a group or individually.

The majority of interventions were delivered in-person (*n* = 15, 48.4%), with 12 delivered in a hybrid in-person and telehealth format (38.7%). Three interventions were delivered entirely via telehealth, and one study gave participants the option of completing the intervention in-person or via telehealth. Of those that delivered the intervention in a hybrid format, the most common telehealth method used was telephone calls (*n* = 11, 91.7%), and one utilized a smartphone app.

The interventions were most commonly administered by nutrition-qualified professionals, including dietitians (*n* = 19, 61.3%) and nutritionists (*n* = 5, 16.1%). Four interventions were administered by non-nutrition-trained researchers, and two studies did not report who administered the intervention. One intervention was delivered entirely online via an app developed by dietitians and did not require intervention administration staff.

### 3.4. Behaviour Change Theories and Techniques

Ten Behaviour Change Technique Taxonomy categories and nineteen techniques were reported across the thirty-one interventions (Table 3). The mean number of techniques was 5.1 and ranged from 2 to 9. Definitions of the identified behaviour change techniques are included in Appendix A. The most common Behaviour Change Technique Taxonomy categories were ‘4.0 Shaping knowledge’ (*n* = 31), ‘3.0 Social support’ (*n* = 24), ‘9.0 Comparison of outcomes’ (*n* = 16), ‘5.0 Natural consequences’ (*n* = 15), ’12.0 Antecedents’ (*n* = 12), and ‘1.0 Goals and planning’ (*n* = 11).

The most common techniques were ‘4.1 Instruction on how to perform the behaviour’ (*n* = 31), ‘3.1 Social support (unspecified)’ (*n* = 24), ‘9.1 Credible source’ (*n* = 16), ‘5.1 Information about health consequences’ (*n* = 15), ’12.5 Adding objects to the environment’ (*n* = 12), and ‘1.1 Goal setting (behaviour)’ (*n* = 9). Only one study directly referenced the BCTTv1 in regard to intervention development.

Of the 31 interventions, the majority did not report using any behaviour change theories in the development of the intervention (*n* = 29 interventions, 93.5%, Table 3). The two interventions that reported the use of theory utilized the social cognitive theory and cognitive learning theory. Four interventions were based on motivational interviewing (12.9%).

## 4. Discussion

Behaviour change theory provides a foundation to explain how and why individuals adopt a behaviour. The findings of this review demonstrate that behaviour change techniques are commonly reported in Mediterranean diet interventions for older adults, with instructions on how to perform the behaviour, social support, and providing information from a credible source being the most common techniques used. Despite the use of behaviour change techniques within interventions, only one study directly referred to using the BCTTv1 when developing the intervention. Indeed, behaviour change theory was rarely described, with the use of theory limited to two studies, one describing the social cognitive theory and one describing the cognitive learning theory.

This review highlights a lack of theoretical underpinning within Mediterranean diet interventions in older adults. The use of theory provides a systematic approach to the design and development of a complex intervention, providing a range of ways to target behaviour change through behaviour change techniques [52]. Only 19 techniques were reported across the included interventions, suggesting that 80% of the available behaviour change techniques are not being utilized, which is considerably less than the techniques that have been used in behaviour change research in other health behaviour interventions [53]. The application of behaviour change techniques has the potential to accelerate behaviour change science but is yet to be realized in this field. This finding underscores the need for clinicians and researchers to utilize an established behaviour change technique taxonomy—such as the BCTTv1—when designing, developing, and reporting on interventions. This will allow for an understanding of the exact nature and content of the intervention, thereby enabling future replication in research, translation into public health practice, and evidence synthesis [54].

The most common component of the included interventions is the use of education and training to increase knowledge and/or understanding of the Mediterranean diet and impart necessary skills to facilitate dietary behaviour change. Studies have shown that the use of education and training in interventions is effective for facilitating change in various lifestyle behaviours, including smoking cessation [55], physical activity [56], and nutrition [57,58]. Knowledge and understanding are key components of behaviour and, importantly, are directly linked to better diet [59]. In Australia, a lack of knowledge about the Mediterranean diet is recognized as a barrier to adoption and adherence, and it has been suggested that having more knowledge would make acceptance of the diet easier [60]. A recent study of Australian adults demonstrated that knowledge of core Mediterranean diet foods, meals, and cooking methods is lacking [61]. However, the evidence for whether education and, in turn, improved knowledge are sufficient for meaningful and sustained behaviour change is mixed [38,41]. The ‘COM-B system’ of behaviour details how an individual’s capability, motivation, and opportunity interact to produce behaviour change [62]. Capability is defined as both the physical and psychological capacities to engage in a behaviour, which include having the required knowledge and skills [60]. The model acknowledges that knowledge alone is not enough to change behaviour, but that change is not possible without it. Indeed, improving knowledge is not inherently linked to changing behaviour [63]. Considering this model, it is likely that differences in diet adherence post-education intervention are due to other behavioural aspects, such as motivations (e.g., addressing a specific diagnosed health disorder, reducing a future risk of illness) and opportunity (e.g., accessibility, affordability).

The findings of the review show that most interventions to date have been delivered in a direct, in-person format. While several studies utilized hybrid in-person and telehealth formats, only three studies included delivering the intervention solely using digital means. The COVID-19 pandemic forced the healthcare system to adapt and illustrated how health services can successfully be provided without the need for in-person contact [64]. Digital health technologies allow for the efficient dissemination of health programs and interventions to individuals in rural and remote regions, without access to transportation, and with limited mobility. They have been utilized in the management and treatment of a range of health conditions, including diabetes [65], obesity [66], sleep problems [67], and depression and anxiety [68]. There is also an increasing interest in the use of digital technologies for self-management and prevention, particularly where time is a significant barrier for clinicians implementing preventative healthcare [69]. Importantly, there is growing evidence to suggest that web-based nutrition education interventions can be effective for increasing awareness, improving knowledge, and improving adherence to the Mediterranean diet [70]. Not only are digital health technologies becoming more accessible—with 93% of Australian older adults having internet access in their homes in 2020—they are also highly sought after by older adults [71,72], although few have been designed specifically for this population. Those that have been applied in the older adult population have been shown to be effective at increasing physical activity [73] and promoting healthy eating behaviours [74].

Importantly, this review identified a large gap in the availability of self-directed, online interventions targeting nutrition—specifically the Mediterranean diet—for older adults. An app-based virtual health coach intervention was successful in increasing Mediterranean diet adherence and was shown to be efficacious and feasible in a sample of older adults, suggesting the utility of digital health technologies for providing education and promoting adherence to the Mediterranean diet [39]. Another study found that dietary counselling provided by a dietitian through a smartphone app led to increased adherence to the Mediterranean diet, comparable to traditional in-person counselling [27]. The Maintain Your Brain trial is a web-based multidomain intervention targeting modifiable risk factors for dementia [75]. The fully online nutrition module promotes the adoption of the Mediterranean diet and is administered to participants who report dietary intakes that do not align with the Mediterranean diet or those who have chronic disease risk factors and may benefit from dietary change. The findings from this trial are expected to be available in 2023 and are expected to provide further evidence regarding the value of digital health technologies for delivering dietary interventions and facilitating behaviour change in older adults [75].

Much of the research in the health and lifestyle space consists of developing strategies and interventions and evaluating their efficacy on clinical markers and hard endpoints. Despite the investment put into designing, developing, and testing the efficacy and internal validity of interventions, there continues to be little or no consideration of the translation of research interventions and their implementation into the general population. Participant characteristics, resource availability, competing time demands, and level of skill of those implementing the intervention are significantly different in the traditional efficacy research environment, compared to real-world settings [76]. This is significant, given that approximately 80% of older adults are living with a chronic condition, most of which require long-term therapy and permanent lifestyle behaviour change [1]. Whilst the Australian Dietary Guidelines provide evidence-based advice on eating habits to promote overall health and reduce the risk of diet-related disease, it is obvious that they are yet to be effective on a population level, given that CVD continues to be a leading cause of death in the country [77]. Current patterns of diet consumption in Australia are marked by high intakes of red meat and low intakes of fruits and vegetables, fish, and legumes, all of which are hallmark components of the Mediterranean diet [11]. Although high Mediterranean diet adherence is possible in Australia, long-term maintenance has been shown to be difficult [78]. During interventions, adherence is achieved using a range of intensive intervention strategies, including one-on-one counselling, supply of food, written resources, and cooking classes [79]. However, this level of interaction and involvement is unlikely to be feasible for the general population, given the lack of specialist nutrition services available, particularly in rural, remote, and disadvantaged areas [80]. The RE-AIM (reach, effectiveness, adoption, implementation, maintenance) framework provides a practical means to enhance the effects of health promotion interventions by evaluating the aspects considered to be the most relevant to real-world implementation [81]. Whilst it has been applied in some settings to evaluate various health behaviour interventions [82,83], it is not routinely utilized, and only one of the studies included in this review reported the use of this framework [49]. If the goal of health promotion is to achieve equity in health, health promotion research should draw on frameworks such as RE-AIM, in conjunction with behaviour change theories and techniques, to ensure that the positive health effects of interventions can be realized by all individuals and not just those who have the resources and opportunities to engage in research [81].

This scoping review has limitations worth noting. The authors cannot exclusively say that Mediterranean diet interventions for older adults were not informed by one or more behaviour change theories; rather, publications did not report the use of the theories for intervention development. The coding of behaviour change techniques depended on reported content. In some instances, descriptions of the interventions were limited and lacked precision. It is possible that the absence of a behavioural strategy could reflect a reporting omission rather than an absence within the intervention. Every effort was made to obtain the study protocol and any additional publications arising from the study to ensure the complete collection of information. Additionally, our assumption that the reporting of education as a behaviour change strategy would involve (at minimum) information about health consequences and instructions on how to perform the behaviour may have resulted in these techniques being over-reported. The alternative, which we felt would have been unhelpful, was to miss participant education in some instances. Only one study that was included reported using the BCTTv1 when developing the intervention; therefore, all behaviour change techniques were subjectively coded. To enhance accuracy and reliability, the coder completed the online BCTTv1 training prior to coding the interventions.

## 5. Conclusions

This scoping review highlights that behaviour change theory is not consistently reported in Mediterranean diet interventions for older adults. Similarly, whilst some behaviour change techniques are used to facilitate behaviour change within interventions, almost 80% of the available techniques are not being utilized. Integrating behaviour change theories and techniques in the development and reporting of nutrition interventions for older adults is essential for effectively targeting behaviours and allowing future replication, translation, and synthesis.

## Figures and Tables

**Figure 1 nutrients-15-01189-f001:**
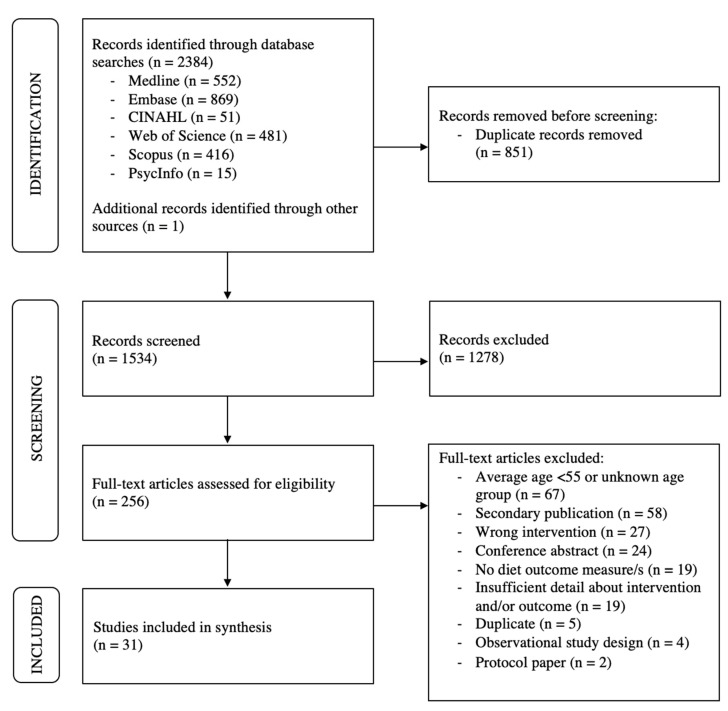
Preferred Reporting Items for Systematic Review and Meta-Analyses Extension for Scoping Review (PRISMA-ScR) of included articles [19].

**Table 1 nutrients-15-01189-t001:** Search strategy for the scoping review of Mediterranean diet interventions among older adults: database—Ovid MEDLINE.

Ovid MEDLINE(R) ALL <1946 to 16 August 2022>
1 Diet, Mediterranean/ 4816 2 Mediterranean diet*.mp. 7055 3 Mediterranean dietary pattern*.mp. 541 4 Cretan diet*.mp. 12 5 Anti-inflammatory diet*.mp. 350 6 MedDiet*.mp. 537 7 MeDi.mp. 1985 8 MeDiet.mp. 45 9 Greek diet.mp. 31 10 1 or 2 or 3 or 4 or 5 or 6 or 7 or 8 or 9 10,121 11 older adult*.mp. 109,936 12 senior*.mp. 49,175 13 Elderly.mp. or Aged/ 3,451,491 14 11 or 12 or 13 3,509,706 15 10 and 14 2468 16 Clinical Trials as Topic/ or Randomized Controlled Trials as Topic/ 354,379 17 intervention.mp. 786,076 18 diet intervention.mp. 1084 19 quasi-experimental*.mp. 17,988 20 16 or 17 or 18 or 19 1,099,171 21 10 and 14 and 20 552

Note: * indicates truncation to return variations on spelling.

**Table 2 nutrients-15-01189-t002:** Inclusion and exclusion criteria of selected studies.

Parameter	Inclusion Criteria	Exclusion Criteria
Population	Community-dwelling older adults with an average age ≥ 55 years	Average age < 55 years Unspecified age group
Morbidities	With or without morbidities	N/A
Study design	Parallel randomized controlled trial Quasi-experimental/pre post	Crossover intervention (not designed to assess behaviour change) Observational (e.g., cross-sectional) Systematic review or meta-analyses
Intervention	Diet only or combined lifestyle (e.g., diet and physical activity) Mediterranean or anti-inflammatory diet Length ≥ 4 weeks Examining: - Adherence to Mediterranean or anti-inflammatory diet - Impact of Mediterranean or anti-inflammatory diet on nutritional behaviours - Impact of Mediterranean or anti-inflammatory diet on nutrition-related health outcomes	Other dietary intervention (e.g., paleo, vegetarian) Non-food-based studies of dietary supplements Laboratory feeding trials
Outcomes	Diet adherence (index score, food, or nutrient intake) Anthropometric outcomes (e.g., blood pressure, weight) Biochemical outcomes (e.g., blood lipids, serum nutrient status)	N/A
Language	English	All other languages

**Table 3 nutrients-15-01189-t003:** Reported behaviour change theories and behaviour change techniques according to the Behaviour Change Technique Taxonomy (version 1) [16] in the included interventions.

	Alonso-Dominiguez et al. [23]	Baguley et al. [24]	Berendsen et al. [25]	Bihuniak et al. [26]	Choi et al. [27]	Chou et al. [28]	Cooper et al. [29]	Davis et al. [13]	Droste et al. [30]	Entwistle et al. [31]	Estruch et al. [32]	Grimaldi et al. [33]	Hagfors et al. [34]	Hardman et al. [35]	Katsarou et al. [36]	Keyserling et al. [37]	Landaeta-Diaz et al. [38]	Maher et al. [39]	Marcos-Forniol et al. [40]	Martinez-Rodriguez et al. [41]	Masumi et al. [42]	Mayr et al. [14]	McGrattan et al. [43]	Michalsen et al. [44]	Quintana-Navarro et al. [45]	Rusch et al. [46]	Salas-Salvado et al. [47]	Schwartz et al. [48]	Toobert et al. [49]	Tuttle et al. [50]	Zuniga et al. [51]
*Behaviour change theory*																															
Social cognitive theory																													●		
Cognitive learning theory						●																									
*Behaviour change technique*																															
1. Goals and planning																															
1.1 Goal setting (behaviour)			●								●					●		●					●		●	●		●			●
1.2 Problem solving							●				●					●		●					●				●				●
1.3 Goal setting (outcome)																											●				
1.5 Review behaviour goal(s)			●								●					●		●					●		●	●		●			●
1.6 Discrepancy between current behaviour and goal			●																						●						
1.7 Review outcome goal(s)																											●				
1.8 Behaviour contract											●																				
2. Feedback and monitoring																															
2.2 Feedback on behaviour	●				●		●											●													
2.3 Self-monitoring of behaviour	●				●			●								●		●					●						●		
2.4 Self-monitoring of outcome(s) of behaviour					●																										
3.0 Social support																															
3.1 Social support (unspecified)	●	●	●	●	●	●	●		●	●	●			●	●	●	●		●				●	●	●	●	●	●	●	●	●
4.0 Shaping knowledge																															
4.1 Instruction on how to perform the behaviour	●	●	●	●	●	●	●	●	●	●	●	●	●	●	●	●	●	●	●	●	●	●	●	●	●	●	●	●	●	●	●
5.0 Natural consequences																															
5.1 Information about health consequences					●	●	●	●		●		●			●					●	●		●	●		●	●	●			●
6.0 Comparison of behaviour																															
6.1 Demonstration of the behaviour				●						●		●	●											●							●
8.0 Repetition and substitution																															
8.1 Behavioural practice/rehearsal					●							●	●								●			●					●		
9.0 Comparison of outcomes																															
9.1 Credible source		●	●	●	●		●	●			●	●	●						●			●	●		●	●	●		●		
10.0 Reward and threat																															
10.1 Material incentive (behaviour)																													●		
10.3 Non-specific reward																					●										
12.0 Antecedents																															
12.5 Adding objects to the environment			●	●			●	●			●		●	●		●						●	●		●		●				

Note: ● = Present in intervention.

## Data Availability

De-identified data from this study are not available in a public archive. De-identified data from this study will be made available (as allowable according to institutional IRB standards) by emailing the corresponding author.

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
