# Peer review of "Behaviour Change Techniques Used in Mediterranean Diet Interventions for Older Adults: A Systematic Scoping Review"

_nutrients, 2023, doi:10.3390/nu15051189_

Round 1

Reviewer 1 Report

This review want to identify and describe behaviour change techniques used to inform Mediterranean diet interventions for older adults. Tha paper is well written and the methods are adequate.

Why the authors identified older adults with >55 years? I suggest to revised this range, because the older adults usually are people >65 years.

Table 1, 2, 3 are not so clear so I suggest to revise. Figure 1 is to revise too.

Why the authors mentioned only the social cognitive theory, cognitive learning theory and transtheoretical model? Why the authors never talk about cogntive behaviotal therapy? 

I suggest to well describe these theory/model in the introduction.

I suggest also to week describe the strategies/tecniques reported.

Reviewer 2 Report

The authors are to be congratulated for their high-quality systematic review which summarizes and evaluates the behavior change strategies used in Mediterranean diet intervention studies in older adults. This is an interesting scoping review which refers to much previous work over a large time-scale The manuscript is well written and easy to follow. I would like to see the following minor issues to be addressed:

1. In the abstract the main findings and conclusion are missing.

2. In the methods section, the search strategy is missing. Please include the MeSH Keywords.

3. Each study must show the level of evidence.

4. Can the authors explain why the manuscript has been made in older adults only?  Please define the limitations.

5. Mediterranean diet could change by country with different patterns of eating. Please list the countries in each study.

6. Please remove the formatting symbols from the text boxes in Figure 1. 

7. Please adjust the table header row in Table3. 
